# Visualization of Aspalathin in Rooibos (*Aspalathus linearis*) Plant and Herbal Tea Extracts Using Thin-Layer Chromatography

**DOI:** 10.3390/molecules24050938

**Published:** 2019-03-07

**Authors:** Emily Amor Stander, Wesley Williams, Fanie Rautenbach, Marilize Le Roes-Hill, Yamkela Mgwatyu, Jeanine Marnewick, Uljana Hesse

**Affiliations:** 1South African National Bioinformatics Institute (SANBI), University of the Western Cape, Robert Sobukwe Road, Bellville 7535, South Africa; emily.amor.stander@gmail.com (E.A.S.); wesleywt@gmail.com (W.W.); yamkelamgwatyu@gmail.com (Y.M.); 2Oxidative Stress Research Centre, Faculty of Health and Wellness Sciences, Institute of Biomedical and Microbial Biotechnology, Cape Peninsula University of Technology, Symphony Road, Bellville 7535, South Africa; rautenbachf@cput.ac.za (F.R.); MarnewickJ@cput.ac.za (J.M.); 3Biocatalysis and Technical Biology Research Group, Institute of Biomedical and Microbial Biotechnology, Cape Peninsula University of Technology, Symphony Road, Bellville 7535, South Africa; LeRoesM@cput.ac.za; 4Institute for Microbial Biotechnology and Metagenomics, University of the Western Cape, Robert Sobukwe Road, Bellville 7535, South Africa

**Keywords:** Rooibos, *Aspalathus linearis*, aspalathin, thin-layer chromatography, p-anisaldehyde, herbal tea, authentication, marker compound

## Abstract

Aspalathin, the main polyphenol of rooibos (*Aspalathus linearis*), is associated with diverse health promoting properties of the tea. During fermentation, aspalathin is oxidized and concentrations are significantly reduced. Standardized methods for quality control of rooibos products do not investigate aspalathin, since current techniques of aspalathin detection require expensive equipment and expertise. Here, we describe a simple and fast thin-layer chromatography (TLC) method that can reproducibly visualize aspalathin in rooibos herbal tea and plant extracts at a limit of detection (LOD) equal to 178.7 ng and a limit of quantification (LOQ) equal to 541.6 ng. Aspalathin is a rare compound, so far only found in *A. linearis* and its (rare) sister species *A. pendula*. Therefore, aspalathin could serve as a marker compound for authentication and quality control of rooibos products, and the described TLC method represents a cost-effective approach for high-throughput screening of plant and herbal tea extracts.

## 1. Introduction

Rooibos (*Aspalathus linearis* (Burm.f.) R. Dahlgren (Fabaceae)) is a leguminous shrub endemic to the Cape Floristic Region of South Africa. The species complex comprises several distinct growth forms, one of which (the Red type) has been cultivated since the 1930s for the production of rooibos herbal tea [1,2]. This tea is caffeine-free, low in tannins, high in volatile compounds and rich in polyphenols [3]. It is gaining international popularity, as an increasing body of literature provides scientific evidence for diverse health promoting properties of rooibos, including antiaging, anticancer, antispasmodic, antidiabetic and cardioprotective activities [4,5,6,7,8]. Many of these effects have been linked to the wide range of phenolic compounds produced by the plant [9,10]. Aspalathin, a C-glucosyl dihydrochalcone, is one of the major polyphenolic rooibos compounds, contributing 4–12% of the total dry matter of the plant [11,12]. Aspalathin is an extremely rare compound, which has so far only been detected in plants of the *A. linearis* species complex and in two populations of the sister species *A. pendula* [10] (but see [1] with regards to *A. pendula*). *A. pendula* is a rare plant species with a limited geographic distribution, used only by the local communities for making “yellow” tea. Therefore, aspalathin represents a potential marker compound for authentication and quality control of plant material and extracts prepared from *A. linearis*.

Fermentation of rooibos, traditionally performed to develop the characteristic red-brown color and slightly sweet flavor, involves chopping, bruising, “sweating” and subsequent drying of the plant material in the sun. The process is associated with a significant (over 90%) reduction in aspalathin content [11,12], as it is oxidized to isoorientin and orientin [13]. Due to the higher polyphenol content, unfermented (green) rooibos is currently finding increased attention for the production of herbal tea and cosmetic products, and as a functional food ingredient [12,14]. So far, producers of green rooibos herbal tea mostly rely on visual inspection, using the red-brown discoloration of the plant material as an indicator for undesirable polyphenol oxidation and aspalathin degradation [15]. Chemical screening is restricted to analyses of the total polyphenol content and total antioxidant capacity, for which comparatively inexpensive and simple methods have been developed [12]. However, these methods do not allow conclusions on the aspalathin content. Exact aspalathin concentrations can be determined using various technologies such as HPLC, as well as Near Infrared and Raman spectroscopy [15,16], but these methods require specialized equipment and expertise, and are therefore prohibitively expensive at high sample numbers.

Thin layer chromatography (TLC) represents a simple and easily scalable method for the detection of diverse compounds in plant extracts, herbal products and foods [17,18]. When paired with chromatogram densitometry or image analysis for compound quantification, TLC provides a cost-effective alternative to other detection methods (e.g., HPLC, NIS, RS) for high-throughput sample screening [18]. Marker compound screening using TLC-densitometry/image analysis requires the establishment of a protocol that ensures reliable compound separation and visualization. A TLC-method for analysis of flavonoids in tea samples, including rooibos tea, has been reported previously [19], however, aspalathin was not determined in that study. P-anisaldehyde is a universal derivatization agent which can be used to detect a wide range of nucleophilic compounds [20]. When investigating flavonoid fingerprints in *Passiflora* species, Birk et al. (2005) [21] found that p-anisaldehyde-H2SO4 represents a suitable alternative derivatization reagent to the commonly used NP/PEG [22], which has also been applied by Ligor et al. (2008) [19]. Derivatization with p-anisaldehyde-H2SO4 not only produced distinct banding patterns, but also appeared to improve band resolution [21].

Here, we tested p-anisaldehyde for its ability to visualize aspalathin in thin layer chromatograms. We describe a simple, fast, cost-effective, sensitive and reproducible TLC method for the reliable detection of aspalathin in rooibos plant material and herbal tea samples. Aspalathin band purity was verified using HPLC-DAD and UPLC-MS, and TLC detection limits were determined using densitometry.

## 2. Results and Discussion

The species *Aspalathus linearis* currently comprises several diverse growth types [1], including prostrate shrublets (Southern and Northern sprouters); more or less densely branched shrubs (Nardouwsberg type, Grey sprouter, Nieuwoudtville sprouter, Wuppertal type); and erect, slender bushes (Red type, Black type, Tree type). The 34 plants analyzed in this study (Table 1, Appendix A) represent a subset from a total of 109 rooibos ecotypes with representatives from eight *A. linearis* growth types, including 28 commercial plants.

### 2.1. Aspalathin Visualization Using TLC

Figure 1 shows TLC analyses conducted with methanol extracts from two aspalathin-producing commercial rooibos plants (1 and 2) and two wild rooibos plants (3 and 4) that produce only trace amounts of the compound (aspalathin concentrations were verified using HPLC-DAD; Table 1). The TLC banding patterns for plants 1 and 2 are representative for commercial rooibos plants: two purple bands at retention factor (RF) 0.53 ± 0.04 and 0.89 ± 0.03 (bands 1 and 3, respectively), which can vary in intensity; and a prominent orange-brown band at RF 0.46 ± 0.03 (band 4). Other bands, like the third purple band at RF 0.64 ± 0.04 (band 2) and the second orange band at RF 0.48 ± 0.03, appear depending on the plant genotype and TLC running conditions.

The prominent orange-brown band 4 in samples 1 and 2 has the same RF value (0.46) and stains the same color as the aspalathin standard. The methanol extracts from plants 3 and 4 did not produce this band, and HPLC-DAD analysis verified only trace amounts of aspalathin (0.01 and 0.02 g/100 g dw, respectively). To verify whether band 4 is produced by the aspalathin in the extracts from plants 1 and 2, we analyzed methanol extracts of silica scraps from the corresponding region on preparative TLC plates using HPLC-DAD (plants 1 and 2) and UPLC-MS (plant 1, 2, 13). For these samples, the silica scraps were found to contain aspalathin at very high purity (HPLC-DAD: Appendix A; UPLC-MS: Figure 2). For plant 3, that produces only traces of aspalathin, silica scraps from the corresponding region of the preparative TLC plate appeared aspalathin-free (HPLC-DAD: Appendix A).

These analyses confirmed that the orange-brown band at RF 0.46 (band 4) was produced by the aspalathin present in the methanol extracts from the commercial and wild rooibos plants, i.e., that the described TLC method is suitable for visualization of this compound in plant extracts. This is the first report on using p-anisaldehyde for visualization of aspalathin in TLC analyses.

### 2.2. Reproducibility and Sensitivity of the TLC Method

To verify reproducibility, the two TLC plates depicted in Figure 1 were prepared by two different persons. The aspalathin bands were present in each of the eight technical repeats of extracts from plants 1 and 2, and absent in the eight technical repeats of extracts from plants 3 and 4.

To determine the limits of detection (LOD) and quantification (LOQ) for pure aspalathin using the TLC method, we analyzed a sequence of defined aspalathin amounts, ranging from 50 ng to 1000 ng. This standard curve analysis was replicated nine times, and aspalathin standards were reproducibly visualized at 200 ng (Figure 3A; 1 replicate). Calculations conducted on the densitometric data of the nine replicates following the International Conference on Harmonization guidelines Q2 [23] (ICH, 2005) allowed determination of an LOD of 135.9 ng and an LOQ of 411.9 ng aspalathin when using the TLC method. To determine LOD and LOQ for aspalathin concentrations in plant samples, methanol extracts of the aspalathin negative plant number 4 were spiked with the compound and applied onto TLC plates to deliver between 400 ng and 3 µg of aspalathin per spot. This standard curve analysis was also replicated nine times (Figure 3B; 1 replicate). The aspalathin band is easily discernable at concentrations of 1 µg. Densitometric image analysis permits verification of aspalathin at much lower concentrations: LOD = 178.7 ng and LOQ = 541.6 ng aspalathin.

### 2.3. TLC Analysis of Extracts from Wild Rooibos Ecotypes

The TLC method was found suitable for visualization of aspalathin in plant extracts from non-commercial rooibos plants. Figure 4 shows the results for the TLC analyses conducted on 12 wild-growing rooibos ecotypes that represent different *A. linearis* growth types from distant locations in the Cederberg Mountain region (Table 1). Of the 12 samples evaluated by TLC, only six developed the orange-brown band at RF 0.46 (band 4), indicating the presence of aspalathin (Figure 4). For these six plants, HPLC-DAD analyses confirmed substantial aspalathin concentrations, ranging from 1.8 to 5.2 mg/100 g plant dry weight (Table 1). Extracts from the plants 6, 7, 8, 14, 15 and 16 did not produce the aspalathin band. HPLC-DAD analyses revealed that the extracts from plants 7, 8 and 14 had very low levels of aspalathin, and that the amounts loaded on the TLC plate (40.7 ng, 16.0 ng and 5.8 ng aspalathin, respectively) were below the calculated LOD (178.7 ng). The plants 5, 15 and 16 did not produce aspalathin at concentrations measurable using the above HPLC-DAD method. The 12 samples showed a high variability in banding patterns, even between plants of the same growth type (e.g., Black type plants 7, 8, 14 and 13; Northern sprouters 11 and 12). Banding patterns were similar for plants from the same populations for the Black type plants 7 and 8 and for the Tree type plants 15 and 16, but not for the Grey sprouters (plants 3 and 4 did not produce measurable amounts of aspalathin; plant 5 did produce measurable amounts of aspalathin).

Previous studies have shown that all eight described *A. linearis* growth types can produce aspalathin [10], though concentrations can vary substantially between populations and individual plants [1,10]. The samples from Southern sprouters and the Black type plants investigated in these studies contained only traces of aspalathin or appeared aspalathin-free, while most of the samples from Red type and Grey type plants were aspalathin rich. Our results are in line with these findings: three of the four Black type plants produced only trace amounts of aspalathin, and the Grey sprouters were either devoid of this compound or aspalathin rich (plant 5: 5 g/100 g dry weight).

The variability in aspalathin production in wild rooibos plants contrasted with the consistent banding patterns observed with extracts from commercial rooibos plants, where the aspalathin band was always clearly visible (Figure 1, Appendix A). The commercially grown rooibos originates from Red type plants that were selected in the 1930s by Dr. P. le Fras Nortier, who favored a bushy upright growth form (which simplified harvest), high growth rates and distinctive taste [12]. This selection may have favored aspalathin-rich plants, which could explain the more uniform banding patterns and consistent visualization of the aspalathin band in our extracts from commercial rooibos plants. Joubert and de Beer (2011) [12] report on variability in aspalathin contents between 21 commercial rooibos plants and mention significant seasonal effects, but the reported concentrations of this compound were never below 2.33 g aspalathin/100 g plant dry matter, as verified by HPLC-DAD.

### 2.4. TLC Analysis of Extracts from Commercial Rooibos Herbal Tea Samples

Green rooibos herbal tea was first introduced in the 1990s as a project of the Agricultural Research Council of South Africa [12] and gained popularity due to higher flavonoid concentrations (including aspalathin) in comparison to fermented rooibos herbal tea [24,25]. To determine whether differences in aspalathin concentrations between green and fermented rooibos can be visualized with the described TLC method, we sampled and analyzed two commercially available green and six fermented rooibos herbal tea brands. Both green rooibos herbal tea brand samples (Figure 5, lanes A and B) had more prominent and darker aspalathin bands than the five fermented rooibos herbal tea brand samples (Figure 5, lanes C-H). HPLC-DAD confirmed that aspalathin was present at much higher concentrations in the green rooibos herbal tea extracts than in the fermented rooibos herbal tea extracts (Table 2), mirroring the TLC results. Therefore, our method can be used to distinguish between green and fermented rooibos herbal tea.

### 2.5. Summary

In this study, we describe a TLC method that allows the detection of aspalathin in rooibos plant material and rooibos herbal tea samples using p-anisaldehyde-H_2_SO_4_ as a derivatization reagent. It reproducibly visualized aspalathin in commercial rooibos plants and permitted sorting of high and low/zero aspalathin producers among the wild rooibos ecotypes. It could therefore be applied for low-cost high-throughput screening of rooibos ecotypes for targeted plant selection. For plant sample analyses, we used mostly aspalathin-rich leaf material, minimized aspalathin degradation by flash-freezing samples in the field and maintaining them at −80 °C, and maximized compound extraction by powdering the samples. Tea samples were analyzed without such pre-processing steps to reflect possible differences in tea quality (as affected by leaf/stem content and particle size due to different sieve sizes). The TLC method allowed differentiation between green and fermented tea samples and could therefore serve for cost-effective high-throughput quality assessment and authentication of green rooibos herbal teas. Aspalathin concentrations in fermented rooibos herbal tea samples were close to the detection limit of the TLC method. For fermented rooibos herbal tea samples, optimization of the TLC method (such as powdering of the leaf material, increased biomass, extraction procedures) is advisable. Furthermore, the TLC method can be adapted for High Performance TLC, which should improve reproducibility and sensitivity.

## 3. Materials and Methods

### 3.1. Reagents and Materials

All solvents (analytical grade) were acquired from Merck (Kenilworth, NJ, USA). Aspalathin was obtained from Chromadex Chemicals (California, CA, USA), and p-anisaldehyde (4-methoxybenzaldehyde) was procured from Sigma-Aldrich (St. Louis, MO, USA).

### 3.2. Plant and Tea Material

Morphologically distinct rooibos ecotypes, including commercially farmed and wild rooibos plants representing eight *A. linearis* growth forms, were sampled in spring (October/November) in the Clanwilliam (Western Cape), Wupperthal (Western Cape) and Nieuwoudtville (Northern Cape) regions in the Cederberg Mountains, South Africa (Table 1, Appendix A). Up to 100 g of rooibos leaves and stems were collected from four sides of each plant, flash frozen in the field with liquid nitrogen, transported on dry ice, and stored at −80 °C. For the analysis of commercially available teas, different brands of green and fermented rooibos herbal tea were purchased from local supermarkets.

### 3.3. Plant, Rooibos Herbal Tea Extract Preparation

Plant samples were freeze dried and ground to a fine powder using a coffee grinder. Then, 50 mg of plant material was extracted in 5 mL methanol. Tea samples remained unprocessed (i.e., were not ground), and 500 mg was extracted in 5 mL methanol. All samples were rotated at 40 rpm on a Benchmark rotating mixer (Benchmark Scientific, Edison, NJ, USA) at 4 °C for 24 h.

### 3.4. Thin-Layer Chromatography (TLC)

The TLC method was adapted from Ligor et al. (2008) [19]. First, a Twin Trough TLC Chamber for 20 × 10 cm TLC plates (CAMAG, Switzerland) was pre-equilibrated for 30 min with solvent vapors by lining the chamber with filter paper moistened with mobile phase solution (acetone-chloroform-water; 80:20:10; *v*/*v*/*v*). For TLC, 10 µL of each plant sample, 15 µl of each tea sample, and 10 µL of the methanol-dissolved aspalathin standard (100 ng/µL) were applied manually as spots to silica gel 60 F254 20 × 20 cm aluminum plates (Merk) using a 200 µL pipet tip. The addition of 10 µL of formic acid per ml of methanol extract just before the TLC analysis was found to substantially reduce tailing of the samples and the aspalathin standard. TLC plates were developed using the mobile phase solution described above. For derivatization, the TLC plates were sprayed until saturation with a p-anisaldehyde solution (0.5% p-anisaldehyde, 1% sulfuric acid (95% to 99%) and 98.5% glacial acetic acid; *v*/*v*/*v*) using an air compressor with air brush (Aircraft Pneumatic System, AS18); and subsequently exposed to heat using a heat-gun (1750 W, Black and Decker) until the bands became visible. RF values for selected bands were calculated using TLC results from 15 plant samples analyzed on different TLC plates. The retention factor (RF) was calculated as the ratio of the travel distance the band to the travel distance of the solvent front. 

Preparative TLC analyses were performed following the above procedures, using 50 µL of plant extract per 3 cm TLC plate. After development, the silica in the region of the aspalathin standard was scraped off and extracted with 1 mL of methanol by vortexing for 20 min at room temperature. The resulting methanol extract was subsequently filtered through a 0.22 µm filter.

### 3.5. Image Analysis for Determination of LOD and LOQ

The TLC plates were scanned using an I-Sensys MF41010 Scanner (Canon) to obtain chromatographic images and densitometry of images was conducted using ImageJ [26].

To determine limits of detection (LOD) and quantitation (LOQ), we conducted TLC analyses with methanol-dissolved commercial aspalathin (200 ng, 300 ng, 400 ng, 500 ng and 1000 ng aspalathin per spot) as well as with aspalathin-spiked plant extracts from an aspalathin-negative rooibos ecotype (400 ng, 500 ng, 1 µg, 2 µg and 3 µg aspalathin per spot). The densitometric measurements were used to calculate LOD and LOQ following the International Conference on Harmonization (ICH) guidelines Q2 (ICH, 2005). Statistical analyses were performed in Excel (Microsoft) using the following formulas: LOD = 3.3 × (standard deviation of the response)/(slope of calibration curve); LOQ = 10 × (standard deviation of the response)/(slope of calibration curve).

### 3.6. High Performance Liquid Chromatography (HPLC-DAD)

HPLC analysis of rooibos methanol extracts was adapted from Bramati et al. (2002) [27] using the Agilent 1200 series HPLC system coupled to a diode array detector (DAD) (Santa, Clara, CA, USA) with a Nucleosil 120-5C18 column 15 cm × 4.6 mm (5µM, Sigma Aldrich). Column temperature was maintained at 21 °C. Mobile phase A was 300 µL/L trifluoracetic acid in water and mobile phase B was 300 µL/L trifluoracetic acid in methanol. A sample volume of 20 µL was injected. A constant flow rate of 1 mL/min was maintained and the gradient elution was performed as follows: 5% B for 5 min, linear increase to 80% B over 20 min, decrease to 35% B over 3 min, 35% B for 2 min, re-equilibration to 5% B. Acquisition was set at 287 nm for aspalathin detection. Peak areas of the samples were compared to a 20 ng/µL aspalathin standard to determine aspalathin concentrations of the samples.

### 3.7. UPLC-MS Analysis

To confirm the purity of the aspalathin peaks obtained using HPLC-DAD, selected samples were also analyzed using high precision UPLC-MS. This analysis was performed at the Central Analytical Facility of Stellenbosch University (WC, South Africa) using a Waters Acquity ultra performance liquid chromatograph (UPLC) coupled to a photo diode array and a Waters Synapt G2 Quadrupole time-of-flight (QTOF) mass spectrometer (MS) equipped with an electrospray ionization source (Waters, Milford, MA, USA). Electrospray ionization was used in negative mode with a cone voltage of 15 V, desolvation temperature of 275 °C, desolvation gas (nitrogen) at 650 L/hr, and the rest of the MS settings optimized for best resolution and sensitivity. Data, acquired in MSE mode, included a low collision energy scan (6 V) from *m*/*z* 150 to 1500; and a high collision energy scan from *m*/*z* 40 to 1500. The high collision energy scan was performed using a collision energy ramp (30 to 60 V). The photo diode array detector was set to scan from 220 to 600 nm. Sodium formate was used for calibration and leucin encephalin was infused in the background as lock mass for accurate mass determinations. Separation was achieved on a Waters HSS T3, 2.1 × 100 mm, 1.7 µm column at a column temperature of 60 °C and a constant flow rate of 0.25 mL/min. An injection volume of 3 µl was used. Mobile phase A was 0.1% formic acid in water, and mobile phase B was acetonitrile. The gradient elution was performed as follows: 100% A for 1 min, linear increase of B to 50% over 21 min, increase of B to 100% over 1 min, wash step at 100% B for 1.5 min, and re-equilibration to initial conditions for 4.5 min. The mass and concentration of aspalathin contained in the samples were compared to an aspalathin standard and an aspalathin standard curve (10 mg/L, 100 mg/L, 200 mg/L and 500 mg/L).

## Figures and Tables

**Figure 1 molecules-24-00938-f001:**
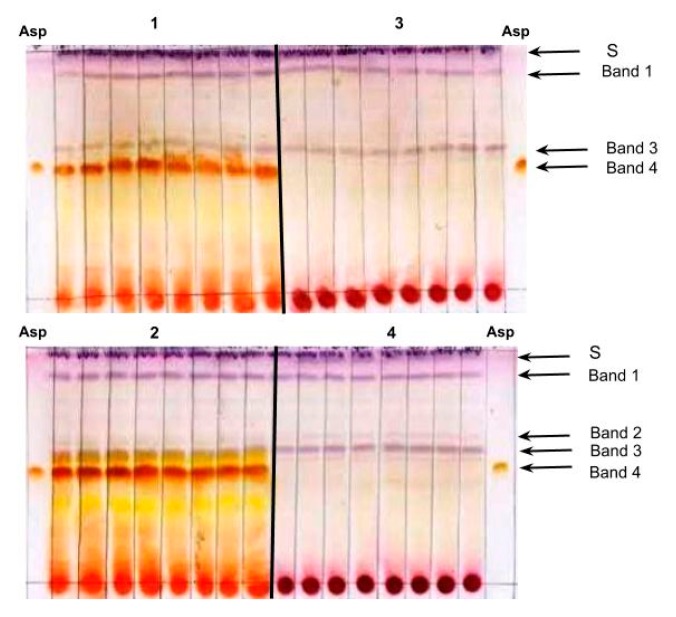
Repeatability test for the TLC method using four rooibos ecotypes with eight repeats. Plants 1 and 2 produce aspalathin, while plants 3 and 4 only produce trace amounts of this compound. Asp: aspalathin standard, S: solvent front.

**Figure 2 molecules-24-00938-f002:**
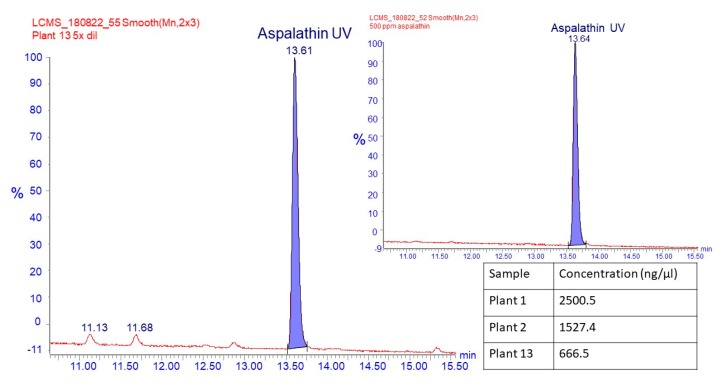
The chromatogram is the UPLC-MS chromatogram comparing TLC silica extracts of plants 1, 2 and 13 at RF = 0.46 to the aspalathin standard (insert). The table shows the concentrations of aspalathin extracted by the preparative TLC extraction method.

**Figure 3 molecules-24-00938-f003:**
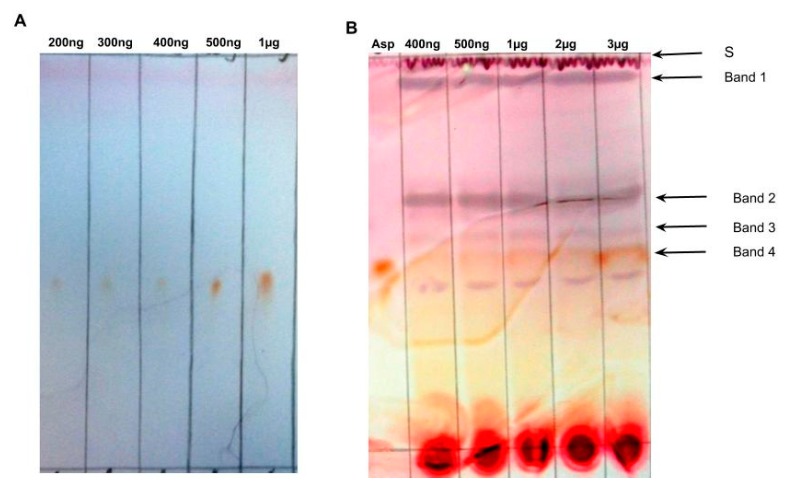
Lowest limit of detection for (**A**) pure aspalathin (*n* = 9, 1 replicate shown), and (**B**) plant 3 (contains no aspalathin) spiked with different aspalathin concentrations (*n* = 9, 1 replicate shown), using TLC. Asp: aspalathin standard, S: solvent front.

**Figure 4 molecules-24-00938-f004:**
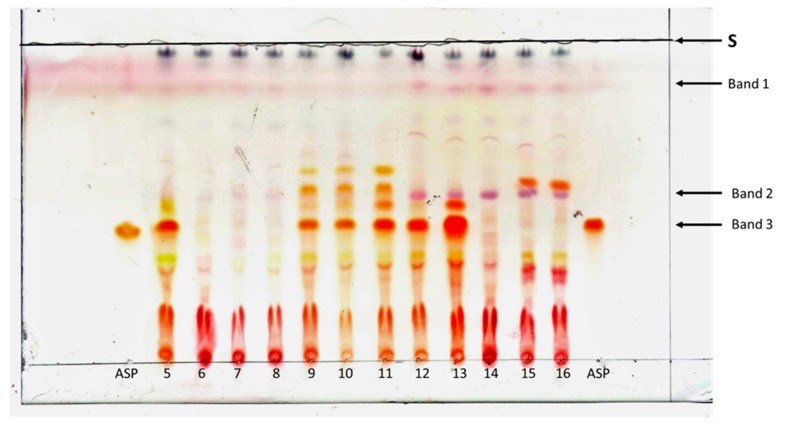
TLC analyses conducted on 12 wild-growing rooibos ecotypes representing different *A. linearis* growth types. An acidifier was included which significantly reduced tailing of the bands. Asp: aspalathin standard, S: solvent front.

**Figure 5 molecules-24-00938-f005:**
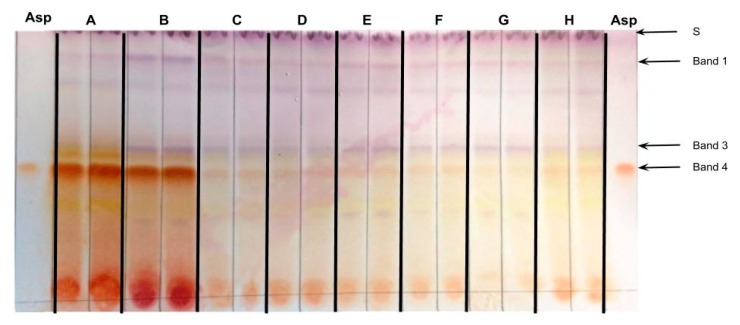
TLC analysis of samples from A) two green (lanes A and B) and six fermented (lanes C–H) rooibos herbal tea brands. Analysis was performed in duplicate. Asp: aspalathin standard, S: solvent front.

**Table 1 molecules-24-00938-t001:** Description of rooibos ecotypes used in the analysis.

Plant	Growth Type	Location	Region	Aspalathin Concentration Determined by HPLC-DAD (g/100 g Plant Dry Weight)
1	Commercial rooibos plant	S032° 47′ 22″ E018° 48′ 26″	Clanwilliam	7.36
2	Commercial rooibos plant	S031° 43′ 17″ E019° 07′ 29″	Nieuwoudtville	12.83
3	Grey sprouter	S032° 37′ 17″ E019° 03′ 24″	Clanwilliam	0.01
4	Grey sprouter	S032° 37′ 17″ E019° 03′ 24″	Clanwilliam	0.02
5	Grey sprouter	S032° 37′ 17″ E019° 03′ 24″	Clanwilliam	4.96
6	Nardowsberg type	S031° 59′ 20″ E018° 50′ 36″	Clanwilliam	0.00
7	Black type	S032° 24′ 38″ E019° 00′ 51″	Algeria	0.04
8	Black type	S032° 11′ 55″ E018° 51′ 43″	Algeria	0.02
9	Nieuwoudtville sprouter	S031° 42′ 55″ E019° 07′ 40″	Nieuwoudtville	1.76
10	Wypperthal type	S032° 20′ 54″ E019° 14′ 26″	Clanwilliam	2.18
11	Northern sprouter	S032° 13′ 29″ E019° 07′ 52″	Clanwilliam	2.72
12	Northern sprouter	S032° 13′ 32″ E019° 07′ 55″	Clanwilliam	1.96
13	Black type	S031° 45′ 48″ E019° 07′ 54″	Nieuwoudtville	5.18
14	Black type	S031° 59′ 21″ E018° 50′ 35″	Clanwilliam	0.01
15	Tree type	S032° 17′ 23″ E018° 51′ 59″	Clanwilliam	0.00
16	Tree type	S032° 17′ 23″ E018° 51′ 59″	Clanwilliam	0.00

Classification of wild rooibos growth types was verified by Prof Van Wyk (University of Johannesburg).

**Table 2 molecules-24-00938-t002:** Aspalathin concentrations (g/100 g plant dry weight) determined by HPLC-DAD in green and fermented rooibos herbal tea extracts.

Tea	Herbal Tea Type	Aspalathin Concentration (g/100 g Tea Leaves)
A	Green	14.47
B	Green	14.20
C	Fermented	0.20
D	Fermented	0.51
E	Fermented	0.50
F	Fermented	1.04
G	Fermented	0.84
H	Fermented	0.99

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
