# Peer review of "Visualization of Aspalathin in Rooibos (*Aspalathus linearis*) Plant and Herbal Tea Extracts Using Thin-Layer Chromatography"

_molecules, 2019, doi:10.3390/molecules24050938_

Round 1

Reviewer 1 Report

The manuscript „Visualization of aspalathin in rooibos (Aspalathus  linearis) plant and herbal tea extracts using thin layer chromatography“ describes the developing and validation of the novel TLC assay for aspalathin detection in rooibos samples.

The described TLC method was simple and fast with a limit of detection for aspalathin below 180 ng and a limit of quantification equal to 541.6ng. The presented methodology might be readily considered as a cost-effective approach for high throughput screening of plant and herbal tea extracts. The manuscript is well-written, performed experiments are logical, obtained results are fully discussed. The manuscript meets all requirements and definitely will be interesting for the audience of the Molecules.

Minor remarks:

1)      Figure 2 is not readable: DAD-response signal (A-C) and MS (D) as well as retention times of the targeted compounds must be bigger and better illustrated.

Author Response

Minor remarks:

1)      Figure 2 is not readable: DAD-response signal (A-C) and MS (D) as well as retention times of the targeted compounds must be bigger and better illustrated.

Reply: To improve Figure 2 in the manuscript, the Figures 2 A, B, C have been moved to the Supplemental section; scales and illustrations have been improved. Figure 2 D is now Figure 2; scales and illustrations have been improved.

Reviewer 2 Report

The manuscript is well present and is recommend for publication after some minor revision.

L107 since authors claim it’s first report on using p-anisaldehyde for visualization of aspalathin in TLC analyses, more information should be provide, such as, how the reaction work, reaction yield, why the authors choose this derivatization agent, selectivity of the reaction, among others.

Other minor issues

L3 space between “tea       extracts”

L5 same  “ Rautenbach 2,                Marilize”

L79 something missing in the middle “bands at RF 0.53         0.04 and 0.89       0.03 (bands”

L81 same “RF 0.64       0.04 (band 2) and the second orange band at RF 0.48       0.03”

Figure 2 please provide figure 2 with better quality, at least the numbers should be visible

from L87 to L179 all text is not format

Author Response

L107 since authors claim it’s first report on using p-anisaldehyde for visualization of aspalathin in TLC analyses, more information should be provide, such as, how the reaction work, reaction yield, why the authors choose this derivatization agent, selectivity of the reaction, among others.

Reply: The following paragraph was added to the Introduction:

Thin layer chromatography (TLC) represents a simple and easily scalable method for detection of diverse compounds in plant extracts, herbal products and foods [Mohammad et al, 2010; Sherma and Rabel, 2018]. When paired with chromatogram densitometry or image analysis for compound quantification, TLC provides a cost-effective alternative to other detection methods (e.g. HPLC, NIS, RS) for high-throughput sample screening [Sherma and Rabel, 2018]. Marker compound screening using TLC-densitometry/image analysis requires the establishment of a protocol that ensures reliable compound separation and visualization. A TLC-method for analysis of flavonoids in tea samples, including rooibos tea, had been reported previously [Ligor et al., 2008], however, aspalathin was not determined in that study. P-anisaldehyde is a universal derivatization agent which can be used to detect a wide range of nucleophilic compounds [Jork et al., 1990]. When investigating flavonoid fingerprints in Passiflora species, Birk et al. (2007) found that p-anisaldehyde-H2SO4 represents a suitable alternative derivatization reagent to the commonly used NP/PEG [Fried and Sherma, 1996], which had also been applied by Ligor et al. (2008). Derivatization with p-anisaldehyde-H2SO4 not only produced distinct banding patterns, it also appeared to improve band resolution [Birk et al., 2007].

Here, we tested p-anisaldehyde for its ability to visualize aspalathin in thin layer chromatograms. We describe a simple, fast, cost-effective, sensitive and reproducible TLC method for reliable separation and detection of aspalathin in rooibos plant material and herbal tea samples.

Added References:

Mohammad, A., S. A. Bhawani, and S. Sharma. "Analysis of herbal products by thin-layer chromatography: a review." International Journal of Pharma and Bio Sciences 1, no. 2 (2010).

Sherma, Joseph, and Fred Rabel. "A review of thin layer chromatography methods for determination of authenticity of foods and dietary supplements." Journal of Liquid Chromatography & Related Technologies 41, no. 10 (2018): 645-657.

Jork, H.; Funk, W.; Fischer, W.; Wimmer, H. Thin-Layer Chromatography Reagents and Detection Methods, VCH Publishers: New York, USA, 1990; Volume la, pp. 195 -198.

Fried, Bernard, and Joseph Sherma. Practical thin-layer chromatography: a multidisciplinary approach. CRC Press, 1996.

Birk, Cristian D., Gustavo Provensi, Grace Gosmann, Flávio H. Reginatto, and Eloir P. Schenkel. "TLC fingerprint of flavonoids and saponins from Passiflora species." Journal of liquid chromatography & related technologies 28, no. 14 (2005): 2285-2291.

Other minor issues

L3 space between “tea       extracts”

Reply: Done

L5 same  “ Rautenbach 2,                Marilize”

Reply: Done

L79 something missing in the middle “bands at RF 0.53         0.04 and 0.89       0.03 (bands”

Reply: Done (added sign for plus minus, which must have disappeared during formatting)

L81 same “RF 0.64       0.04 (band 2) and the second orange band at RF 0.48       0.03”

Reply: Done (added sign for plus minus, which must have disappeared during formatting)

Figure 2 please provide figure 2 with better quality, at least the numbers should be visible

Reply: To improve Figure 2 in the manuscript, the Figures 2 A, B, C have been moved to the Supplemental section; scales and illustrations have been improved. Figure 2 D is now Figure 2; scales and illustrations have been improved.

from L87 to L179 all text is not format

Reply: Text has been reformated

Reviewer 3 Report

This is an interesting investigation that may have a significant impact on analyzing aspalathin in a great variety of rooibos at low cost. However, authors should add information about advantages and disadvantages of tested methods (HPLC-DAD or UPLC-MS versus TLC), as well as the factors that may conduct to overestimated results by TLC such as TLC conditions, concentration of aspalathin in rooibos, the chemistry and size of the particles, thickness of the layer, among others. TLC method is a “qualitative” analysis method, that coupled with other methods, such as the densitometry or mass spectrometry, can be considered as a quantitative one. So, this information must be clear in the document. 

Other important point that authors must considered is the reorganization of the article, material and methods sections must be added before results and discussion. Additionally, the experiment design and the statistical analysis of the results must be added.

 Abstract and introduction: Overall, these sections are clear, however, author should specify that thin layer chromatography is less sensitive for determining aspalathin than HPLC and Near Infrared and Raman spectroscopy.

Line 117: what is the meaning of ICH?  

Table 1. Please add a description of the plants A to H in a table footnote.

Authors should add the material and methods sections in point 2 instead of point 3.

There is not shown the experiment design and the statistical analysis of the results.

Lines 272-273: Are these conclusions obtained from your own experiment? These statements must be included in the introduction.

Conclusion: This section must be rewritten in basis of your own results, showing their importance.

Author Response

Abstract and introduction: Overall, these sections are clear, however, author should specify that thin layer chromatography is less sensitive for determining aspalathin than HPLC and Near Infrared and Raman spectroscopy.

Reply: In this study, we have not compared sensitivity of HPLC-DAD, NIS, RS and TLC with regards to aspalathin amount determination, and therefore refrain from stating that our TLC method is less sensitive. We have clarified our aims in the Introduction as follows:

Here, we tested p-anisaldehyde for its ability to visualize aspalathin in thin layer chromatograms. We describe a simple, fast, cost-effective, sensitive and reproducible TLC method for reliable detection of aspalathin in rooibos plant material and herbal tea samples. Aspalathin band purity was verified using HPLC-DAD and UPLC-MS, and TLC detection limits were determined using densitometry.

Authors should add information about advantages and disadvantages of tested methods (HPLC-DAD or UPLC-MS versus TLC). TLC method is a “qualitative” analysis method, that coupled with other methods, such as the densitometry or mass spectrometry, can be considered as a quantitative one. So, this information must be clear in the document. 

Reply: The following paragraph was added to the Introduction:

Thin layer chromatography (TLC) represents a simple and easily scalable method for detection of diverse compounds in plant extracts, herbal products and foods [Mohammad et al, 2010; Sherma and Rabel, 2018]. When paired with chromatogram densitometry or image analysis for compound quantification, TLC provides a cost-effective alternative to other detection methods (e.g. HPLC, NIS, RS) for high-throughput sample screening [Sherma and Rabel, 2018]. Marker compound screening using TLC-densitometry/image analysis requires the establishment of a protocol that ensures reliable compound separation and visualization. A TLC-method for analysis of flavonoids in tea samples, including rooibos tea, had been reported previously [Ligor et al., 2008], however, aspalathin was not determined in that study. P-anisaldehyde is a universal derivatization agent which can be used to detect a wide range of nucleophilic compounds [Jork et al., 1990]. When investigating flavonoid fingerprints in Passiflora species, Birk et al. (2007) found that p-anisaldehyde-H2SO4 represents a suitable alternative derivatization reagent to the commonly used NP/PEG [Fried and Sherma, 1996], which had also been applied by Ligor et al. (2008). Derivatization with p-anisaldehyde-H2SO4 not only produced distinct banding patterns, it also appeared to improve band resolution [Birk et al., 2007].

Authors should add information about the factors that may conduct to overestimated results by TLC such as TLC conditions, concentration of aspalathin in rooibos, the chemistry and size of the particles, thickness of the layer, among others.

Reply: The following paragraph was added to the Results and Discussion section:

2.5 Summary

In this study we describe a TLC method that allows detection of aspalathin in rooibos plant material and rooibos herbal tea samples using p-anisaldehyde-H2SO4 as a derivatization reagent. It reproducibly visualized aspalathin in commercial rooibos plants, and permitted sorting of high and low/zero aspalathin producers among the wild rooibos ecotypes. It could therefore be applied for low-cost high-throughput screening of rooibos ecotypes for targeted plant selection. For plant sample analyses, we used mostly aspalathin-rich leaf material, minimized aspalathin degradation by flash-freezing samples in the field and maintaining them at -80˚C, and maximized compound extraction by powdering the samples. Tea samples were analysed without such pre-processing steps to reflect possible differences in tea quality (as affected by leaf/stem content and particle size due to different sieve sizes). The TLC method allowed differentiation between green and fermented tea samples, and could therefore serve for cost-effective high-throughput quality assessment and authentication of green rooibos herbal teas. Aspalathin concentrations in fermented rooibos herbal tea samples were close to the detection limit of the TLC method. For these samples optimization of the TLC method (such as powdering of the leaf material, increased biomass, extraction procedures) is advisable. Furthermore, the TLC method can be adapted for High Performance TLC, which should improve reproducibility and sensitivity.

Line 117: what is the meaning of ICH?  

Reply: Inserted “International Conference for Harmonization (ICH)”

The following Reference was added:

ICH, ICoH. "Q2 (R1): Validation of analytical procedures: text and methodology." In International Conference on Harmonization, Geneva. 2005.

Table 1. Please add a description of the plants A to H in a table footnote.

Reply: Table 1 provides descriptions for 16 of the 34 investigated plants, including growth types and plant location. The designations A-H are provided in Table 2 and refer to rooibos tea samples. We have corrected the column header in Table 2.

Other important point that authors must considered is the reorganization of the article, material and methods sections must be added before results and discussion. Authors should add the material and methods sections in point 2 instead of point 3.

Reply: Please refer to “Molecules” instructions for authors, which require that the Methods section is inserted after the Results section.  https://www.mdpi.com/journal/molecules/instructions

Additionally, the experiment design and the statistical analysis of the results must be added.

Reply: We have clarified the aim of our study to explain absence of experimental design and statistical analyses.

1.   In this study, we modified a previously published TLC method (Ligor et al., 2008) by substituting the derivatization reagent NP/PEG with p-anisaldehyde-H2SO4 to test if we can visualize aspalathin. We did not optimize TLC parameters, since the aim (visualization of aspalathin using thin layer chromatography) has been met. We therefore do not describe experimental design and did not conduct statistical analyses.

2.   HPLC-DAD and UPLC-MS were only used to confirm presence/absence of pure aspalathin in the TLC band. Therefore, statistical analyses were not conducted to compare sensitivity of the different methods.

The following paragraph was changed in the Introduction:

Here, we tested p-anisaldehyde for its ability to visualize aspalathin in thin layer chromatograms. We describe a simple, fast, cost-effective, sensitive and reproducible TLC method for reliable detection of aspalathin in rooibos plant material and herbal tea samples. Aspalathin band purity was verified using HPLC-DAD and UPLC-MS, and TLC detection limits were determined using densitometry.

Lines 272-273: Are these conclusions obtained from your own experiment? These statements must be included in the introduction. Conclusions: This section must be rewritten in basis of your own results, showing their importance

Reply: The Conclusions have been removed and the following summary has been added to the Results and Discussion section that reflects our results

2.5 Summary

In this study we describe a TLC method that allows detection of aspalathin in rooibos plant material and rooibos herbal tea samples using p-anisaldehyde-H2SO4 as a derivatization reagent. It reproducibly visualized aspalathin in commercial rooibos plants, and permitted sorting of high and low/zero aspalathin producers among the wild rooibos ecotypes. It could therefore be applied for low-cost high-throughput screening of rooibos ecotypes for targeted plant selection. For plant sample analyses, we used mostly aspalathin-rich leaf material, minimized aspalathin degradation by flash-freezing samples in the field and maintaining them at -80˚C, and maximized compound extraction by powdering the samples. Tea samples were analysed without such pre-processing steps to reflect possible differences in tea quality (as affected by leaf/stem content and particle size due to different sieve sizes). The TLC method allowed differentiation between green and fermented tea samples, and could therefore serve for cost-effective high-throughput quality assessment and authentication of green rooibos herbal teas. Aspalathin concentrations in fermented rooibos herbal tea samples were close to the detection limit of the TLC method. For these samples optimization of the TLC method (such as powdering of the leaf material, increased biomass, extraction procedures) is advisable. Furthermore, the TLC method can be adapted for High Performance TLC, which should improve reproducibility and sensitivity.

Reviewer 4 Report

Comments regarding the manuscript presented and the following should be amended:

Line 35: give full botanical designation of the plant. Aspalathus linearis (Burm.f.) R. Dahlgren (Fabaceae)

Line 36: The species complex comprises------

Line 117: densiometric should read densitometric

Lines 192-199: season of collection should be mentioned

Lines 200-204: IF this method were to be adopted as a means of assessment of quality - there should be further specifications of ‘powder’ with regard to sieves used etc.

Since Rooibos tea is prepared from the cut stems and leaves of Aspalathus linearis (Burm.f.) R. Dahlgren (Fabaceae). The highly lignified stems which make up about 45% of the product, will contain less of the active constituents. This explains why it is important to regulate the powder size which in turn is controlled by the size of sieves used during the powdering process. This also reduces the variability of constituent analysis [Kotina et al., 2012]. Joubert and de Beer [2012; Joubert et al., 2012] also refer to sampling industry practice of using different sieve sizes to create the different qualities of rooibos tea (A, B, C and D).

Line 210: applied manually- ie. as spots or bands?

Line 221: underivatized silicon layer --- should be silica

REFERENCES

Kotina E.L., et al., 2012. The pharmacognostic value of leaf and stem anatomy in rooibos tea (Aspalathus linearis). South African Journal of Botany 82:129-133

Joubert E, de Beer D. 2012. Phenolic content and antioxidant activity of rooibos food ingredient extracts. Journal of Food Composition and Analysis 27:45-51

Joubert E., et al., 2012. Variation in phenolic content and antioxidant activity of fermented herbal tea infusions: Role of production season and quality grade. Journal of Agricultural and Food Chemistry 60:9171-9179

Author Response

Line 35: give full botanical designation of the plant. Aspalathus linearis (Burm.f.) R. Dahlgren (Fabaceae)

Reply: Done

Line 36: The species complex comprises------

Reply: Done

Line 117: densiometric should read densitometric

Reply: Done

Lines 192-199: season of collection should be mentioned

Reply: Done

Lines 200-204: IF this method were to be adopted as a means of assessment of quality - there should be further specifications of ‘powder’ with regard to sieves used etc.

Since Rooibos tea is prepared from the cut stems and leaves of Aspalathus linearis (Burm.f.) R. Dahlgren (Fabaceae). The highly lignified stems which make up about 45% of the product, will contain less of the active constituents. This explains why it is important to regulate the powder size which in turn is controlled by the size of sieves used during the powdering process. This also reduces the variability of constituent analysis [Kotina et al., 2012]. Joubert and de Beer [2012; Joubert et al., 2012] also refer to sampling industry practice of using different sieve sizes to create the different qualities of rooibos tea (A, B, C and D).

Reply: the powdering of the plant samples and absence of preprocessing of tea samples is now explained in the Materials and methods section. In addition, the effect of particle size on TLC method performance is discussed in the Results and Discussion section as follows:

2.5 Summary

In this study we describe a TLC method that allows detection of aspalathin in rooibos plant material and rooibos herbal tea samples using p-anisaldehyde-H2SO4 as a derivatization reagent. It reproducibly visualized aspalathin in commercial rooibos plants, and permitted sorting of high and low/zero aspalathin producers among the wild rooibos ecotypes. It could therefore be applied for low-cost high-throughput screening of rooibos ecotypes for targeted plant selection. For plant sample analyses, we used mostly aspalathin-rich leaf material, minimized aspalathin degradation by flash-freezing samples in the field and maintaining them at -80˚C, and maximized compound extraction by powdering the samples. Tea samples were analysed without such pre-processing steps to reflect possible differences in tea quality (as affected by leaf/stem content and particle size due to different sieve sizes). The TLC method allowed differentiation between green and fermented tea samples, and could therefore serve for cost-effective high-throughput quality assessment and authentication of green rooibos herbal teas. Aspalathin concentrations in fermented rooibos herbal tea samples were close to the detection limit of the TLC method. For these samples optimization of the TLC method (such as powdering of the leaf material, increased biomass, extraction procedures) is advisable. Furthermore, the TLC method can be adapted for High Performance TLC, which should improve reproducibility and sensitivity.

Line 210: applied manually- ie. as spots or bands?

Reply: Done, as spots

Line 221: underivatized silicon layer --- should be silica

Reply: Done

Round 2

Reviewer 3 Report

This is an interesting investigation that may have a significant impact on analyzing aspalathin from rooibos at low cost. 

The manuscript has been improved, now it can be considered for publication in this Journal.